# Gradient boosted decision trees reveal nuances of auditory discrimination behavior

**Carla S. Griffiths[1], Jules M. Lebert[1], Joseph Sollini[1,2], Jennifer K. Bizley**  **[1] \***

**1** Ear Institute, University College London, London, United Kingdom, **2** Hearing Sciences, University of Nottingham, Nottingham, United Kingdom

\* j.bizley@ucl.ac.uk

## Abstract

Animal psychophysics can generate rich behavioral datasets, often comprised of many 1000s of trials for an individual subject. Gradient-boosted models are a promising machine learning approach for analyzing such data, partly due to the tools that allow users to gain insight into how the model makes predictions. We trained ferrets to report a target word's presence, timing, and lateralization within a stream of consecutively presented non-target words. To assess the animals' ability to generalize across pitch, we manipulated the fundamental frequency (F0) of the speech stimuli across trials, and to assess the contribution of pitch to streaming, we roved the F0 from word token to token. We then implemented gradient-boosted regression and decision trees on the trial outcome and reaction time data to understand the behavioral factors behind the ferrets' decision-making. We visualized model contributions by implementing SHAPs feature importance and partial dependency plots. While ferrets could accurately perform the task across all pitch-shifted conditions, our models reveal subtle effects of shifting F0 on performance, with within-trial pitch shifting elevating false alarms and extending reaction times. Our models identified a subset of non-target words that animals commonly false alarmed to. Follow-up analysis demonstrated that the spectrotemporal similarity of target and non-target words rather than similarity in duration or amplitude waveform was the strongest predictor of the likelihood of false alarming. Finally, we compared the results with those obtained with traditional mixed effects models, revealing equivalent or better performance for the gradient-boosted models over these approaches.

## Author summary

The sorts of listening challenges faced by real-world listeners are rarely captured by most laboratory-based auditory paradigms, particularly those testing animal models. However, many labs are attempting to utilize more realistic experiments, and more complicated behavioral paradigms require more sophisticated approaches to analyzing the resulting data. Here, we used a new behavioral paradigm to test the ability of ferret listeners to identify target speech sounds and assess their ability to generalize across changes in pitch. To make sense of the resulting dataset, we used machine learning to understand how trained ferrets perform this task. Gradient-boosted regression and decision trees are well-

**Data Availability Statement:** All data from this publication are freely available on figshare (https://doi.org/10.5522/04/25386565.v1). All code to run the analysis within the paper are available on

GitHub (https://github.com/carlacodes/boostmodels).

**Funding:** This work was supported in part by a Wellcome Trust/Royal Society Sir Henry Dale Fellowship Grant 098418/Z/12/A to JKB, a European Research Council Consolidator award (SOUNDSCENE) to JKB and a Biotechnology and Biological Sciences Research Council research grant (BB/N001818/1) to JKB. The funders had no role in study design, data collection and analysis, decision to publish, or preparation of the manuscript.

**Competing interests:** The authors have declared that no competing interests exist.

established machine learning methods that do not require users to predetermine interaction effects and are accompanied by visualization tools that allow insights into how multiple factors ultimately shape behavior. We compare the use of gradient-boosted models to more standard regression approaches and that this machine learning approach is ideal for analyzing behavioral data in animal models.

## Introduction

Psychophysics paradigms in non-human animals are often designed to yield tractable datasets for relating brain and behavior. Most common laboratory-based paradigms rely on artificial stimuli presented within the confines of simple tasks—such as two-alternative forced choice paradigms in which animals must discriminate a single sound token, or Go/No-Go tasks in which animals detect a change in a repeating sequence of sounds. Such paradigms offer tight experimental control, and can be successfully analyzed using standard statistical approaches such as mixed effect models and more sophisticated approaches that allow, for example, the identification of how and when non-sensory factors shape performance [1, 2]. Yet animals can be trained to perform more complex tasks, generating rich behavioral datasets that potentially can require new approaches for their interpretation. One promising approach for modeling both categorical and continuous data is gradient-boosted decision and regression trees [3]. Not only are such models powerful, but they are also interpretable through the use of tools that allow visualisation of the contributions of variables and combinations of variables to prediction outcomes.

The general approach of the gradient-boosted decision and regression tree model is a form of ensemble learning in which we use an initial weak decision tree to predict an outcome of a trial and then iteratively build upon the error of the first tree (after calculating the loss) by further splitting the data in a way that improves the model prediction. Once our loss plateaus or we reach the maximum number of training epochs, we stop training the model and calculate our test accuracy, or how well the model could predict our target variable on a held-out test set of data. We chose this method as our data is inherently dense (from long periods of behavioral training and testing) and tabular, which makes gradient-boosted regression and decision trees an excellent candidate for the prediction of binary data (such as was the trial a hit or a miss) and continuous data (such as reaction times) compared to a nonlinear neural-network-based classifier [3]. Here, we highlight the utility of both the model itself and the visualization tools available to understand what features the model finds informative and compare this approach to more traditional mixed effects models.

We applied gradient-boosted models to animal psychoacoustics data designed to probe the role of pitch in perceptual invariance and auditory scene analysis. Pitch is a fundamental feature of a person's voice, and a hallmark of human voice processing is recognizing a word regardless of voice pitch. Differences in pitch allow us to separate competing voices, while sounds are grouped together over time into 'streams' if they share a common pitch [4]. However, it is not clear whether the ability to use pitch continuity to link sounds into streams is uniquely human or whether it can be considered a more general feature of the mammalian auditory system. To address such issues, we trained ferrets to detect the word "instruments" within a stream of other randomly drawn non-target words (Sollini and Bizley, in prep.). Within a trial, all word tokens were drawn from a single female or male voice, and the whole stream could be shifted upwards or downwards in fundamental frequency (F0, which determines pitch). The F0 of each word within a stream could also be randomly shifted to assess whether pitch contributes to streaming. We collected 20487 trials of data from 5 animals. We

analyzed these using gradient-boosted models to address two research questions: firstly, can trained ferrets generalize their learned discrimination across variations in pitch, and secondly, whether, like humans, animals can use pitch as a streaming cue to link sounds together over time.

Through the application of gradient-boosted models, we were able to demonstrate that while performance was robust to changes in pitch, shifting the F0 of words within a trial significantly slowed reaction times and elevated the likelihood of a false alarm, providing evidence that ferrets, like humans, use pitch to form perceptual streams. Moreover, this approach allowed us to identify words that ferrets consistently confused with the target word, suggesting that errors were not simply random lapses in attention. Analysis of acoustic features of non-target words identified spectro-temporal similarity but not duration or waveform similarity as a predictor of the likelihood of a false alarm.

## Methods

### Ethics statement

All experimental procedures were approved by local ethical review committees (Animal Welfare and Ethical Review Board, at University College London and the Royal Veterinary College, University of London, and performed under license from the UK Home Office (Project Licenses PP1253968, 70/7267).

**Animals.**   Subjects were five pigmented ferrets (*Mustela putorius*, female) who started training from 6 months of age and were tested until ages between 18 months and 4 years of age. Animals were maintained in groups of 2 or more ferrets in enriched housing conditions, with regular otoscopic examinations to ensure the cleanliness and health of ears. All animals were trained in the behavioral task, using water as a reward. During testing periods, animals were water-regulated. Animals were tested twice daily from Monday to Friday, with free access to water from Friday afternoon to Sunday afternoon. Each ferret received a minimum of 60 ml/kg of water per day through a combination of task performance and supplementation with a wet mash made from water and ground high-protein pellets. Each ferret's weight and water consumption were logged daily throughout the experiment.

**Equipment.**   We controlled the task and stimulus presentation through an RZ6 controller (Tucker Davis Technology, Florida, USA) using OpenEx with custom-written "GoFerret" software [5] on a Windows PC. The right and left-hand speakers were calibrated to match the sound levels using a Bruel & Kjaer measuring amplifier (Type 2610). We presented each trial at a mean sound level of 65 dB SPL; stimuli were scaled to be constant in sound level across trials and talker types.

**Stimuli.**   Stimuli were composed of a sequence (or 'stream') of consecutively presented words, all of which came from the same talker. Continuous speech from two talkers (1 male, 1 female) reading the same passage from the Spoken Corpus Recordings in British English (SCRIBE) database was manually segmented into words and linked together with a minimum gap of 0.08s of silence between words. The audio files were recorded at 20000 Hz but upsampled to 24,414 Hz for presentation.

**Task.**   In a sound discrimination task, we trained five ferrets to recognize the target stimulus (the word 'instruments') against 54 other non-target stimuli (which were also English words) in a stream. Each stream (or string of words) consisted of a series of non-target words and one occurrence of the target word, which could occur anytime from 500 ms to 6.5 s after the onset of the trial (with the target timing drawn from a uniform distribution). As well as being preceded by non-target words, the target was followed by a sequence of non-target words that exceeded the duration of the response time (2s, see below). Streams were

constructed de novo at the start of each trial with non-target words drawn randomly (with replacement) from a pool of 54 non-target words per talker.

The whole trial was comprised of word tokens from the same talker and presented from either the left or right speaker. Once trained, animals were required to initiate a trial by nose-poking at a center port that contained an infrared sensory and water delivery system. They were required to maintain contact until the target was presented. Once the target sound was presented, they were required to move to the response port on the same side as the stimulus presentation. A correct response required the animal to release the center port within 2s of the target word onset and correctly lateralize the sound stream (although, in practice, animals rarely made localization errors). Catch trials (25% of all trials) contained only non-target words and were constructed to be equal in duration to the non-catch trials. On catch trials, the animal was required to remain at the center port and received a water reward from the central port at the end of the trial if they did so.

**Training.**   Initially, ferrets were trained to move between the 3 lick ports (left, center, and right side) by alternating water reward at each port. Once this was accomplished (usually within 1 to 2 sessions), they were trained to lateralize the target sounds ('instruments'). This was achieved by rewarding the initiation of a trial (a response at the center port) and presenting several repetitions of the target sound from one of the lateral locations (either left or right). The ferret would receive a second reward only if they responded at the corresponding location. Once ferrets could perform this target lateralization task at a high rate of performance (>90% correct) over = >2 sessions, the delay between initiating the trial and presenting the target word was systematically increased to 5 seconds over sessions, requiring that performance remained above 80% correct as the delay was increased. Once the ferret was capable of waiting 5 seconds at the center port for target presentation and accurately lateralizing the stimulus, we reduced the target presentation to a single-word token. We then gradually introduced non-target words before and after the target. Non-target words were initially presented with a 60 dB attenuation cue that was gradually reduced until animals were performing with the target and non-target at an equivalent sound level. 3/5 animals were trained first on the female talker and then the male talker, whereas F2105 and F2002 were trained with both simultaneously. All word tokens within a trial were drawn from the same talker, but the talker identity was randomly drawn across trials. Even once trained, we included a proportion of trials (25–50%) that included a 10 -20 dB attenuation cue. These trials were excluded from the analysis but helped maintain the animals' motivation to perform the task. 25% of trials were catch trials in which the target word was not presented. Baseline training varied in duration from 3 months to 8 months.

**Pitch roving.**   Animals were considered fully trained once they consistently performed above 70% correct on trials without an attenuation cue (chance performance is approximately 33% given the 6s trial duration and a 2s response window, i.e., 2s / 6s = 1/3). Once trained on the natural ('Control') F0 trials, we introduced F0 (pitch) roving. For each talker, we used STRAIGHT (which separates source and resonator information, therefore allowing manipulation of F0) [6] to shift the F0 up or down by 0.4 octaves. This resulted in F0 values of 109 and 144 for the male voice, where the natural F0 was 124 Hz, and 144 and 251 Hz for the female voice, where the natural F0 was 191 Hz.

In inter-trial roving, the pitch of the entire trial shifted up or down, whereas, in intra-trial roving, the F0 value of each word was randomized. As in training, all word tokens within a trial came from the same talker.

**Data analysis.**   Any trial has four possible outcomes: hit, correct response, miss, and false alarm. A hit was defined as moving away from the center port ('release') and responding at the target location within 2s of the target word presentation. A correct rejection was defined as

remaining at the central port for the entire duration of the trial (on a catch trial), a miss as failing to leave the central port within 2s of the target word presentation, and a false alarm as releasing from the center port before target word presentation or the end of a catch trial. False alarms immediately terminated the sound presentation and elicited a time-out (signaled by a modulated noise burst). Time outs lasted 2 seconds, during which the ferret could not reinitiate a trial.

We define $p(hit) = n\ hits/(n\ hits + n\ misses)$, and the proportion of false alarms (FA) as $p(FA) = n\ false\ alarms/[n\ hits + n\ misses + n\ correct\ rejections + n\ FA]$. We consider correct responses (C.R.) as either a hit or a correct reject, where $p(correct) = [n\ hits + n\ correct\ rejections]/[n\ hits + n\ misses + correct\ rejections + n\ FA]$. We also calculated a sensitivity metric (d') [7], where $d' = z(p(hits)) - z(p(FA))$, where z represents the normal distribution function. We define reaction time as the central port release time rather than the lateral response time relative to the timing of the target word. To analyze whether word tokens systematically elicited behavioral responses, we defined the response time as the exit time from the central port relative to trial onset. All data analysis, from behavioral metrics to computational models, was programmed using Python 3.9.

**Computational models.**   The general approach of the gradient-boosted decision and regression tree model is a form of ensemble learning in which we use an initial weak decision tree of a depth larger than 1 to predict an outcome of a trial based on our behavioral data and then iteratively build upon the error of the first tree (after calculating the loss) by constructing the next tree based on the residuals of the previous tree. Once the loss plateaus or a maximum number of training epochs is reached, training stops and test accuracy is calculated by assessing how well the model predicts a held-out test set of data. We chose this method as our data is inherently dense (from long periods of behavioral training and testing) and tabular, which makes gradient-boosted regression and decision trees an excellent candidate for the prediction of categorical and continuous data compared to a nonlinear neural-network-based classifier [3].

Linear mixed effect and generalized linear models are commonly used alternatives that allow trial-based analysis of categorical or continuous behavioral data. While powerful, such models can fail to capture non-linear or non-monotonic relationships that might be present in behavioral data. Machine learning approaches offer an alternative model-free approach to uncovering statistical structure in rich behavioral data sets such as those typical of animal behavioral work. Models were generated using LightGBM [8]. Gradient-boosted regression trees were used to model reaction time data. Gradient-boosted decision trees were used to make classification models for binary trial outcomes (hit vs. miss and false alarm vs. correct rejection). To optimize hyperparameters for this model, we implemented a grid search using optuna [9].

We generated 5 models to address our research questions. Two classification models were developed; one considered determining whether a ferret missed a target word (miss vs. hit model), and the second considered the factors that influenced the likelihood of a false alarm/correct rejection of a non-target word (false alarm/correct reject model). Our reaction time model used gradient-boosted regression to determine the parameters influencing the animals' reaction time to the target word. Our response time models (one each for male and female talker trials) predicted the release time within a trial based on the timing of the words. They were used to assess whether animals made systematic false alarms with particular words. Hyperparameters for model fitting are provided in the supplemental tables (S15 and S16 Tables).

We determined which features were significant using cumulative feature importance, which sums the contributions of each variable across all of the trees in which it is utilized, and permutation testing, which shuffles a feature of our data (e.g., the target F0) and then selects the drop in performance the model has due to that feature being shuffled. We generated

permutation importance plots from the sci-kit learn (sklearn) package to quantify the extent to which shuffling any given feature decreased the quality of the model, thereby establishing which features contributed significantly to model performance. The classification models were tuned using binary log loss with an evaluation metric of binary log loss across 10,000 epochs and implemented early stopping of 100 epochs. The regression models implemented the l2 loss function over 1000 epochs with an early stopping of 100 epochs. For the classification models, all hyperparameter optimization minimized binary log loss, whereas, for the regression (reaction time) model, hyperparameter optimization minimized the mean-squared error (l2 loss function).

The regression models' test and train mean-squared error was calculated using 5-fold cross-validation. The train and test accuracy and balanced accuracy were calculated using 5-fold cross-validation for the classification models. Noise floors were calculated for the regression models by calculating model performance when utilizing trials in which the relationship between reaction / response times was randomly shuffled 1000 times, while keeping the hyper-parameters constant. We then used Shapley Additive values to assess parameter influence on the trial outcome. For the classification models, this was the likelihood of a miss/hit or false alarm; for the regression models, this was the reaction time. To visualise the contributions of model features (i.e., feature importance and cumulative feature importance), we used the SHAP package [10], an implementation of Shapely Additive Importance features to elucidate explainability from the typically 'black-box' regression and classification tree models. The SHAP package allowed us to plot partial dependency plots to see how the impact of the model would vary as inter-related features changed (such as talker gender and trial number). For the categorization models, we applied subsampling of the data to equalize trial counts; to force the model to weight trial types with equal importance, we sub-sampled control F0 trials to match intra and inter-F0 roved trials. To weigh the trial outcomes with equal importance, we sub-sampled hit responses to match the number of miss responses for the miss/hit classification model and sub-sampled non-false alarm responses to match the number of false alarm responses in the false alarm model.

For the regression model that calculated the absolute response time within the trial (rather than relative to the target), we used sub-sampling to create a uniform distribution of words. This sub-sampling, or bootstrapping, was done so our gradient-boosted regression tree model wouldn't associate higher-frequency words with a higher likelihood of a false alarm or response just because of its higher frequency. However, word sampling was not always fully independent, particularly for a subset of trials in two animals in which some of the words were programmed to occur 80% more frequently than other words in order to optimise data collection for neural recordings. Thus, to achieve something close computationally to a mathematically perfect bootstrapping procedure, we created a loop for each of the 54 non-target words, found the trials that contained that non-target word, and placed them into a data frame.

We then sub-sampled this resulting data frame to 700 samples (the minimum number of counts across all words in the original data frame) unless the non-target was a naturally high-frequency occurring word, where it was sub-sampled to 50 samples or skipped entirely. After all 54 words were iterated through in order, the resulting sub-sampled data frame was appended to an array. Next, we repeated the same process but went through the non-target words in reverse order to ensure some words wouldn't be over-sampled in the resulting distribution. This whole process of iterating through all the non-target words and flipping the order of iteration was repeated 18 more times (S5(A) and S5(B) Fig). The results reported are from this subsampled data frame. However, we repeated the analysis using the natural (biased) frequencies of word occurrences and obtained very similar results (S5(C) Fig), illustrating that the GBM does not require balanced data to yield sensible results.

We implemented generalised linear mixed effect models and linear mixed effects models using the rpy2 and statsmodels packages in Python using a binomial model (logit link) for categorical data or a gaussian model (identity link) for reaction time data.

## Results

### Ferrets can discriminate speech sounds, and their performance is robust to pitch shifting

Ferrets were trained to detect the target word "instruments" within a stream of randomly drawn non-target word tokens. Subjects initiated a trial by nose-poking in a central port that contained an infrared sensor and water delivery spout and were required to remain at the center port until the presentation of the target word. On each trial, all tokens came from the same talker and position in space, and ferrets were rewarded for responding at the lateral port adjacent to the speaker within 2s of the target word (Fig 1A and 1B). On catch trials, in which only

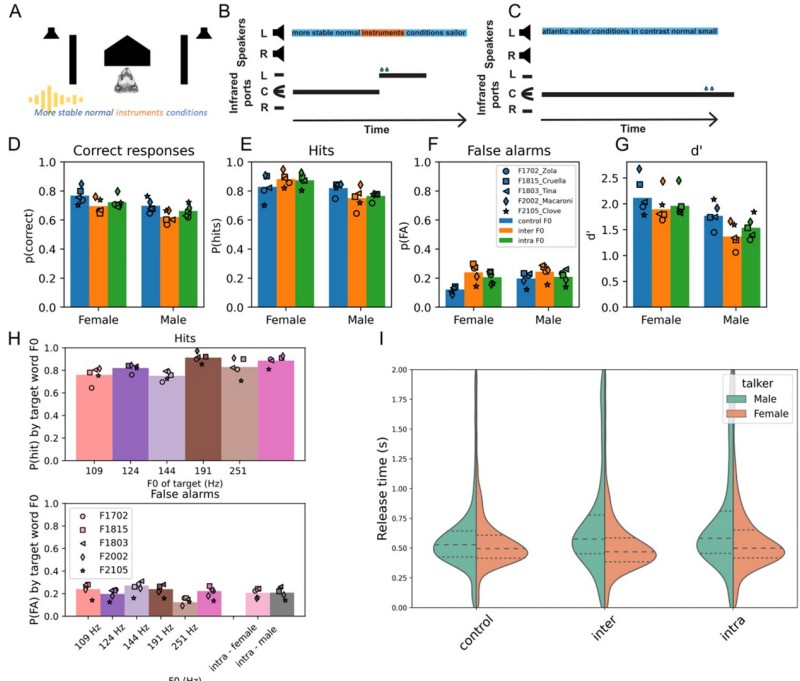

**Fig 1. Task design and basic behavioral data.** A, Schematic of the experimental booth. To trigger a trial, ferrets had to nose-poke a center port that contained an IR sensor and water port. This triggered the presentation of a stream of words from either the left or right speaker (Image source: Journal of Genetics, Vol. XI, No. 2, Public Domain, https://commons.wikimedia.org/wiki/File:Ferret-Polecat-Hybrid.jpg). B, Ferrets were trained to remain at the center until the presentation of the target word ('instruments') and received a water reward at a lateral port if they correctly released within 2s of target presentation and responded to the lateral port whose side matched that of the speech stream. C, Catch trials did not contain the target word, and the ferret was rewarded if she remained at the central port for the duration of the trial. D, Behavioral metrics across animals distributed by talker type. Bars indicate the across-animal average; symbols show the individual animals. Trials are separated according to the identity of the talker and the pitch roving condition (control = no pitch shifting, inter = F0 shifting of the whole trial, intra = F0 shifting of the tokens within a trial). (D) % correct over all trials, E, hits; F, false alarms; G, sensitivity (d'). H, impact of F0 on hit rate (top) and false alarm rate (bottom). False alarm rates are plotted separately for intra-trial pitch roving because the F0 changed from token to token, making it impossible to assign a false alarm to a distractor of a given F0. I, Violin plot of reaction times during correct responses on trials in which the target was correctly identified for all animals, separated by talker type.

non-target words were presented, ferrets were rewarded for remaining at the central port (Fig 1C). Ferrets were trained with a single male and single female voice. Once performance was stable, trials were introduced in which the F0 of the whole trial was shifted ('inter-trial roving') or individual word tokens within the trial were shifted ('intra-trial roving'). We will first provide an overview of the data before using gradient boosted decision trees to understand and quantify the factors that shape the animals' performance in this task.

Ferrets' were able to learn and perform the task across control and F0-shifted conditions; performance ranged from 57%-85% correct for all animals and conditions, where 33% would be considered chance performance (Fig 1D). Hit rates were generally high (Fig 1E) and false alarms low (Fig 1F) for both talkers and both types of pitch-shifted trials. Overall, performance was higher for the female voice, with a small decrease in d' evident for pitch-roved trials compared to natural F0 ones (Fig 1G). Nonetheless, all d' values were well above 1, indicating the animals were well able to perform the task.

To understand whether ferrets form a pitch-tolerant representation of the target word, we considered the impact of F0 changes on performance (Fig 1D–1F). Two-way repeated measures ANOVAs with factors talker (male/female) and rove type (control / inter / intra) showed that for hit rates, there was a significant effect of talker and significant talker x rove interaction, but no significant pairwise comparisons across pitch roved conditions (S1 and S2 Tables). For false alarms, there was again a significant effect of talker, rove, and talker x rove interaction, with posthoc comparison showing that for the female talker control, F0s elicited significantly lower false alarm levels than either rove type but that the rove types were not significantly different from each other (S3 and S4 Tables). For sensitivity (d') measures, there were again significant effects of talker and rove type, but post hoc comparisons showed no rove conditions to be significantly different from each other (S5 and S6 Tables). Therefore, overall, while subjects were better on female talker trials than on male talker trials, the performance on inter and intra-trial roved trials was largely equivalent (Fig 1D–1F). When the performance was broken down according to the actual F0 value, we observed there was a modest influence of F0 on hit rates, such that the highest hit rates were observed for the female talker's up-shifted F0 trials (Fig 1H). False alarms, in contrast, were lower for the control F0 values for both the male and female talkers.

Reaction times varied by ferret and according to the talker (S1(B) Fig). The trend for lower hit rates at lower F0 and for the female voice to elicit faster reaction times may be a consequence of training, as 3/5 subjects were initially trained on only the female talker. However, while the hearing range of ferrets fully encompasses that of humans, their frequency resolution is poorer and most notably so at the lowest audible frequencies [11], and this too may limit performance at the lowest F0s.

These basic behavioral metrics are designed only to show that ferrets can successfully discriminate a target word from non-target words despite variation in F0. We now turn to gradient-boosted models (GBMs) to further consider how acoustic and non-acoustic factors influence individual trial outcomes.

## Introduction to gradient boosted models

Gradient boosting is a supervised machine learning algorithm used for classification and regression problems and is particularly advantageous due to the tools available to visualize how a model exploits information to perform the task. The basic principle is that decision trees are built by splitting observations based on feature values, with the algorithm seeking and selecting a split that results in the highest gain in information by comparing predicted outcomes to observed ones. We chose this machine learning approach as our data is abundant in

sample size and tabular. While its application to animal behavioral work is to our knowledge novel, this scenario of structured, dense data is ideal for gradient-boosted decision trees, as this type of method has often been used in recommender systems [12] as well as economic predictive modeling for human behavior in customer loyalty [13]. A machine learning approach is ideal because it can uncover non-linear dependencies in the data without users being required to predetermine interaction effects in their model. Moreover, we can consider multiple stimulus features, such as the talker and pitch of the word, as well as the trial history parameters (was the previous trial correct, was the previous trial a catch trial) and non-stimulus features (such as the timing of the trial within the session, the time of the target word within the trial, and the side that the animal was required to respond) that may influence performance but do not necessarily inform our experimental hypothesis.

We used lightGBM [8] to implement a gradient-boosted machine (GBM) approach. We considered two types of models—decision-tree models that performed categorical discriminations, for considering whether responses to targets were misses or hits and whether responses to catch trials were false alarms or correct rejections, and regression tree models to predict continuous reaction time data. In each case, we trained models using 5-fold cross-validation and used held-out data to report both the accuracy and balanced accuracy (which is particularly helpful for data in which observations are unequal in number between categories and where accuracy may, therefore, be overinflated). To assess which variables were utilized by the model, we used two metrics; feature importance and permutation importance. The GBM decision and regression tree method consists of many trees, and features will potentially be used many times to split the data; to understand the contribution of a feature, the gain provided must be aggregated across trees. Therefore, the feature importance metric assesses how a given feature improves the model's accuracy by summing the gain provided by that feature across all of the times that it's used in the model. A higher gain implies that the feature is more important for generating predictions. In lightGBM, the loss functions (from which gain is computed) are the mean squared error (MSE) for regression tasks and the log loss for classification tasks. Its units are the same as the target variable, seconds, and its upper and lower bounds are minus to positive infinity. Permutation importance provides a complementary measure of the importance that any given feature provides to the model. The permutation feature importance is the decrease in a model score when a single feature is randomly permuted. The higher the permutation importance, the larger the contribution a variable makes to the model; a score of 0.1 for a model with 70% accuracy reflects a drop to 60% accuracy for a classification problem. One caveat with the permutation importance is that it assumes that all variables are independent, so it can underestimate the contribution of a given variable in some circumstances [14].

To visualise the way in which variables impacted model predictions, and how variables interact with one another we used SHapely Additive exPlanations (SHAPs) which are a common way of understanding machine learning models based on Shapely values. Shapely values were derived from cooperative game theory and represent the average contribution of each feature to all possible combinations of features [10]. SHAPs extend this to machine learning models; for every feature and every observation in the training set, we obtain a SHAP value, and therefore, there are as many SHAP values as there are observations. For a classification task the SHAP values are expressed as the log(odds) so can be directly interpreted as the impact of a given feature on the probability e.g. of making a miss. For our regression models, the SHAP scores are the impact on reaction times, expressed in seconds. Here we use SHAP summary plots to provide intuitive and interpretable visualizations of the effects of all variables in a model and partial dependency plots to visualize combinations of features of interest. The partial dependency plots are particularly helpful for understanding how, for example, behavior

varies across individual subjects and for examining the potentially non-linear interactions between features that the model has learned to exploit.

## Talker identity drives miss responses

We used lightGBM [8] to model the likelihood of a miss vs hit response using only trials in which the target sound was presented(i.e., excluding false alarms and catch trials). The variables provided to the model were: the talker (male/female), the side (left/right) of the audio presentation, the trial number (in the session), the subject identity (ID), target presentation time (within the trial), the target F0, whether the previous trial was a catch trial, whether the previous response was correct, and whether the F0 of the non-target word preceding the target matched that of the target (non-target F0 = target F0, this selects intra-trial roved trials eliminating those trials where by chance the word before the target matched the target F0).

The performance of the miss/hit model was reasonable despite the sparsity of miss responses in the behavioral data, with an average balanced accuracy on our a training set of 62.17% and an average test balanced accuracy of 62.09%. We eliminated factors that either did not significantly increase the cumulative feature importance plot (Fig 2A) or if a permutation test that randomized the variable in question did not impact model fit (Fig 2B). Thus, trial history factors (the past trial was correct or a catch trial) and the preceding non-target F0 = target

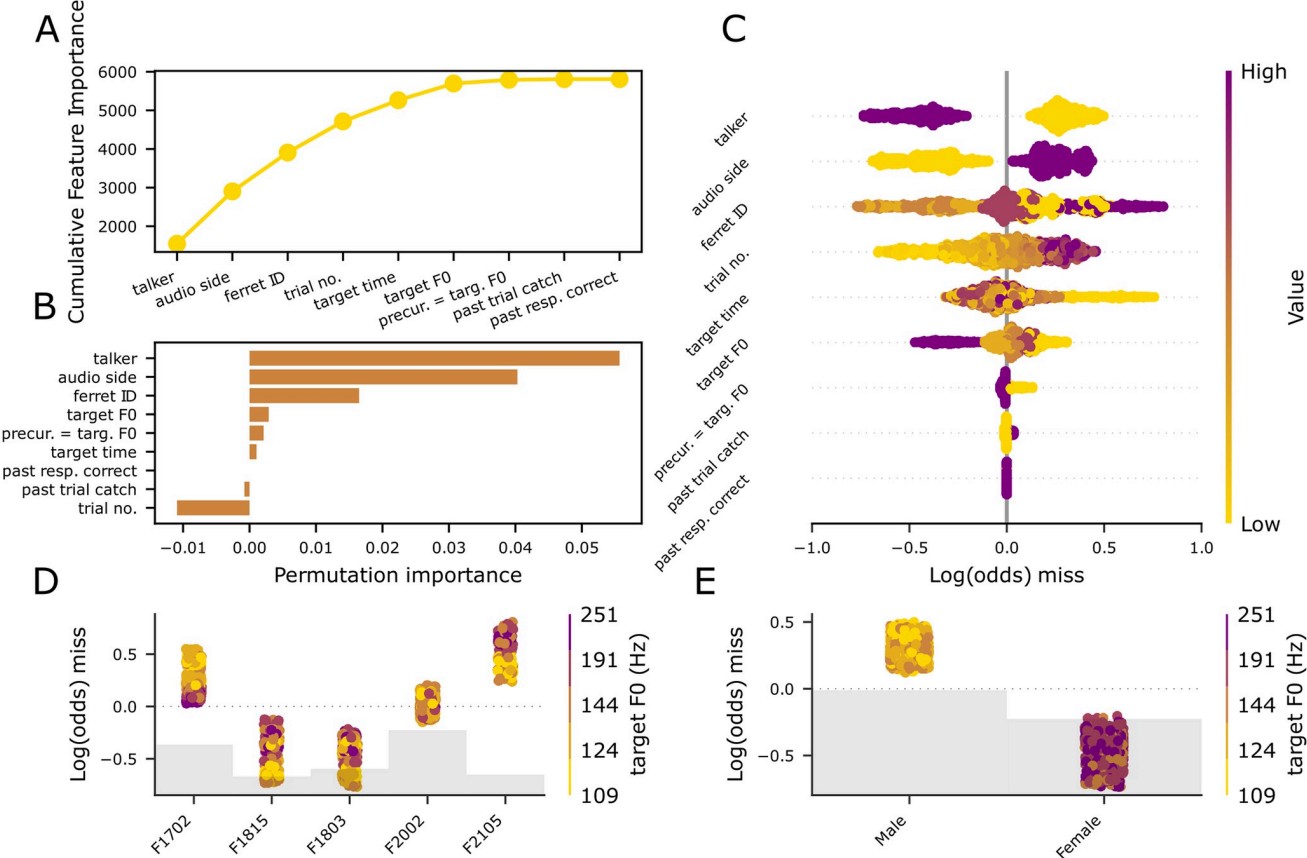

**Fig 2. Talker identity is the strongest predictor of misses.** A, the elbow plot of cumulative feature importance over trial features; B, permutation importance bar plot of the features in the correct hit/miss model; C, SHAP feature importances of the miss/hit model; D, SHAP partial dependency plot depicting the SHAP impact over each ferret ID color-coded by target F0. E, SHAP partial dependency plot showing the SHAP impact over each talker type color-coded by target F0. Gray bars indicate the distribution of the number of observations across variables.

F0 parameter were eliminated. For the remaining features, the feature importance metrics, permutation tests, and SHAP feature values were all in concordance with each other, with only minor differences in the ranking of features. The top three features were the talker (the male talker increased the probability of a miss, Fig 2C), the side of the audio presentation (which was idiosyncratic across animals, likely reflecting their own individual biases, see S2(B) Fig) and the trial number (with trials earlier in the session reducing the likelihood of a miss, and later trials being associated with higher miss rates). The target presentation time within the trial was significant when assessed by feature importance metrics and not when assessed via the permutation test (S2 Fig). In any case, there was not a strong or consistent relationship between miss probability and target time across animals, as shown by the lack of consistent stratification in the SHAPs plot examining the target presentation time impact for each ferret. The F0 of the target sound also had a small but significant effect, which varied by ferret (Fig 2E). Only 3/5 animals had stratified miss probabilities which suggested higher F0s were more likely to elicit false alarms. In contrast, one animal (F1702) showed the opposite pattern the final animal (F2002) showed no consistent pattern. Whether the non-target word that preceded the target word was matched in F0 did not significantly influence the likelihood of missing. We conclude that the talker's identity was the single biggest stimulus factor that altered the likelihood of missing, with the F0 of the target word having a modest effect in some animals. Changing the F0 from word token to word token did not change the likelihood of correctly detecting the target.

## False alarms are influenced by talker identity and F0

Next, we modeled whether a subject would false alarm based on all trial types, using the following features: the talker, the pitch (F0) of the trial or for intra-trial roved trials the F0 of the last non-target word in the trial, the side of audio presentation, the trial duration, the time elapsed since the start of the trial, the trial number within the experimental session, the ferret ID, whether the past response was correct, whether the past trial was a catch trial, and whether there was intra-trial F0 roving. The false alarm model had above-chance accuracy (mean test accuracy of 61.54% over 5-fold cross-validation; balanced accuracy 61.46%) and returned the following as the most significant contributors: the time elapsed since the trial started, the trial number, the ferret ID, the non-target F0, the audio side, and whether the trial was intra-trial F0 roved (Fig 3A, 3B and 3D).

In contrast to the miss model, the strongest determinants of whether an animal was likely to false alarm were timing parameters (time in the trial and trial number within the session) and the individual ferrets. Partial dependency plots (S3 Fig) showed that two ferrets were more likely to false alarm early in the trial, one late in the trial, and two animals showed unstratified responses, implying they were not systematically influenced by this parameter (S3(A) Fig). Trial number, although significant, did also not show clear stratification when considered by animal (S3(H) Fig).

The speech sound F0 and talker both impacted the likelihood of FA, with the partial dependency plot showing that low F0 words spoken by the female talker were most likely to elicit an FA. In contrast, the control F0 for the female talker was least likely to elicit an FA (Fig 3D and S3(C) Fig). The audio side and intra-trial roving also contributed to the model: the audio side was again idiosyncratic across animals (S3(B) Fig). Whether or not word tokens within a trial varied in F0 (i.e., intra-trial roving) contributed a significant effect in the predicted direction (i.e., intra-trial roving was more likely to elicit an FA), but only 3 / 5 ferrets showed this, and overall, it was a small effect (Fig 3E).

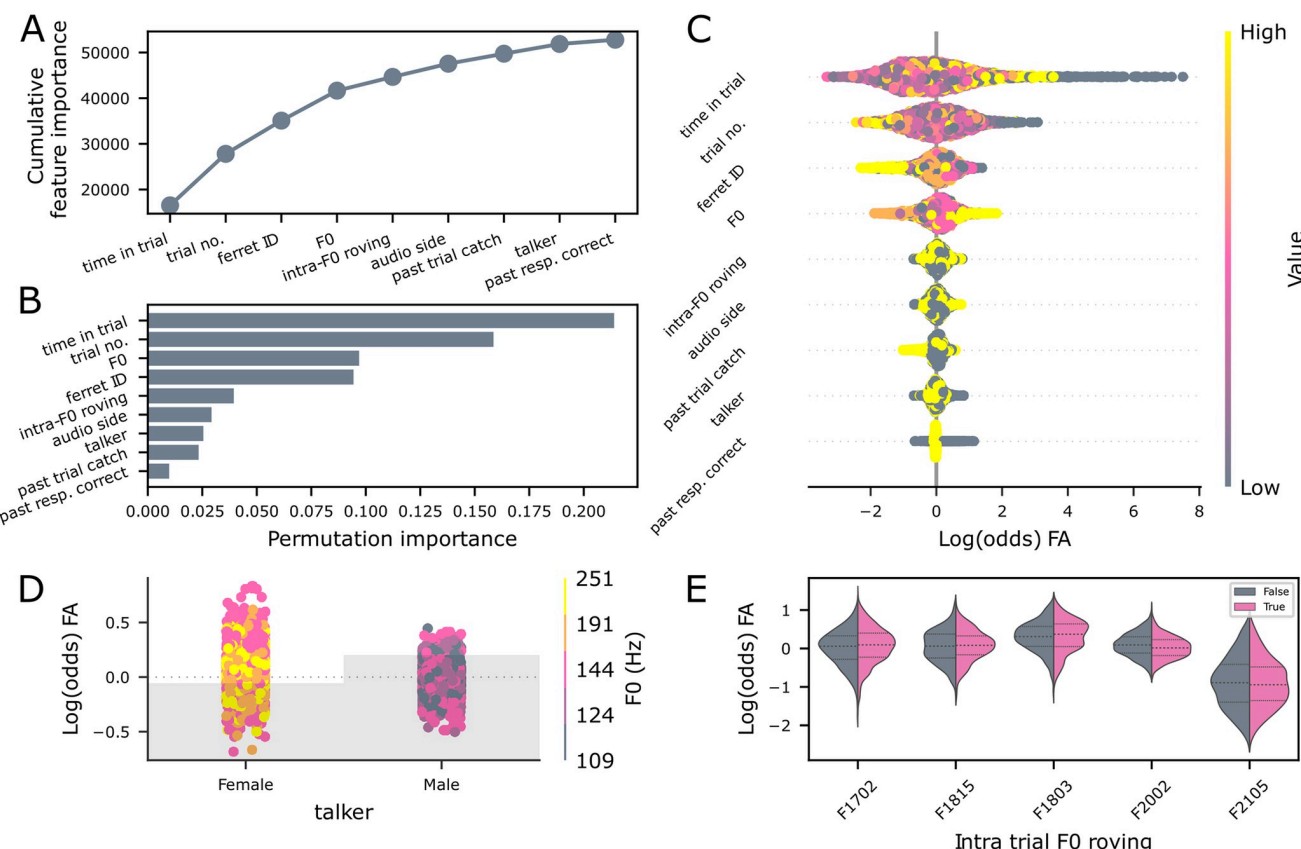

**Fig 3. Acoustic and trial timing factors influence false alarms.** A, elbow plot depicting the cumulative feature importance of each factor used in the false alarm decision tree model; B, Permutation importance plot. C, SHAP feature importance values; D, partial dependency plot depicting the SHAP value over whether the trial was intra-trial roved color-coded by F0. E, partial dependency plot showing the SHAP value (representing the impact on the probability the trial would be predicted as a false alarm) over ferret ID color-coded by whether the trial was intra-trial roved; Gray bars illustrate the relative proportion of trials across categories.

In summary, the FA model suggests that non-acoustic factors are the key drivers in whether animals false alarm with only a small contribution of acoustic factors. Pitch-shifting, particularly within trials, had small but measurable effect on false alarm rate.

## Gradient boosted regression of reaction time data reveals the impact of pitch on target detection and streaming

Given our performance measures were generally quite high with, in particular, a very limited number of miss trials with which to explore whether F0 changes impacted performance, we focused next on reaction time (RT) measures. To explore whether RTs provided a more sensitive measure of how acoustic and task parameters influenced performance, we used gradient-boosted regression [8]. In our RT model, derived from responses from correct non-catch trials, we considered the following factors: ferret ID, talker (male or female), time to target presentation (within a trial), the trial number (within a session), the side of audio presentation, the target F0, whether the F0 changed from the preceding non-target word to the target word (preceding F0 = target word F0), whether the past trial was a catch trial, and whether the past trial was correct. Our test-set mean squared error (mse) using 5-fold cross-validation was 0.102s compared to a noise floor (calculated by randomizing the relationship between trials

and reaction times) test mse of 0.133s (train mean-squared error = 0.092s, compared to a noise floor train mse of 0.105s).

From the permutation test, the ferret ID, the talker, the side of the audio presentation, the time to target presentation, the target F0, and trial number were significant factors (Fig 4B), whereas SHAP values additionally considered whether the F0 of the previous word equaled the target word as a significant factor in this reaction time model (Fig 4A). This difference in traditional permutation importance versus SHAP feature importance is not necessarily surprising, as target F0 is correlated with the precursor = target F0 feature (i.e., if the target F0 is not a control F0, the likelihood of precursor not equalling target F0 increases), something which the permutation importance method struggles to account for [14]. Interestingly, a traditional mixed effects model (see below and Fig 7C) also returned whether the percursor was the same F0 as the target word as a significant variable, with trials in which both shared the same F0 having faster reaction times than those that did not. Similar to the miss/hit and false alarm/correct reject performance models, the model heavily weighted both ferret ID and talker ID; reaction times were longer for the male talker (in 4/5 ferrets, see Supplemental S4D, female slower in F2105) and varied systematically across ferrets (Fig 4C). Overall, later targets had faster

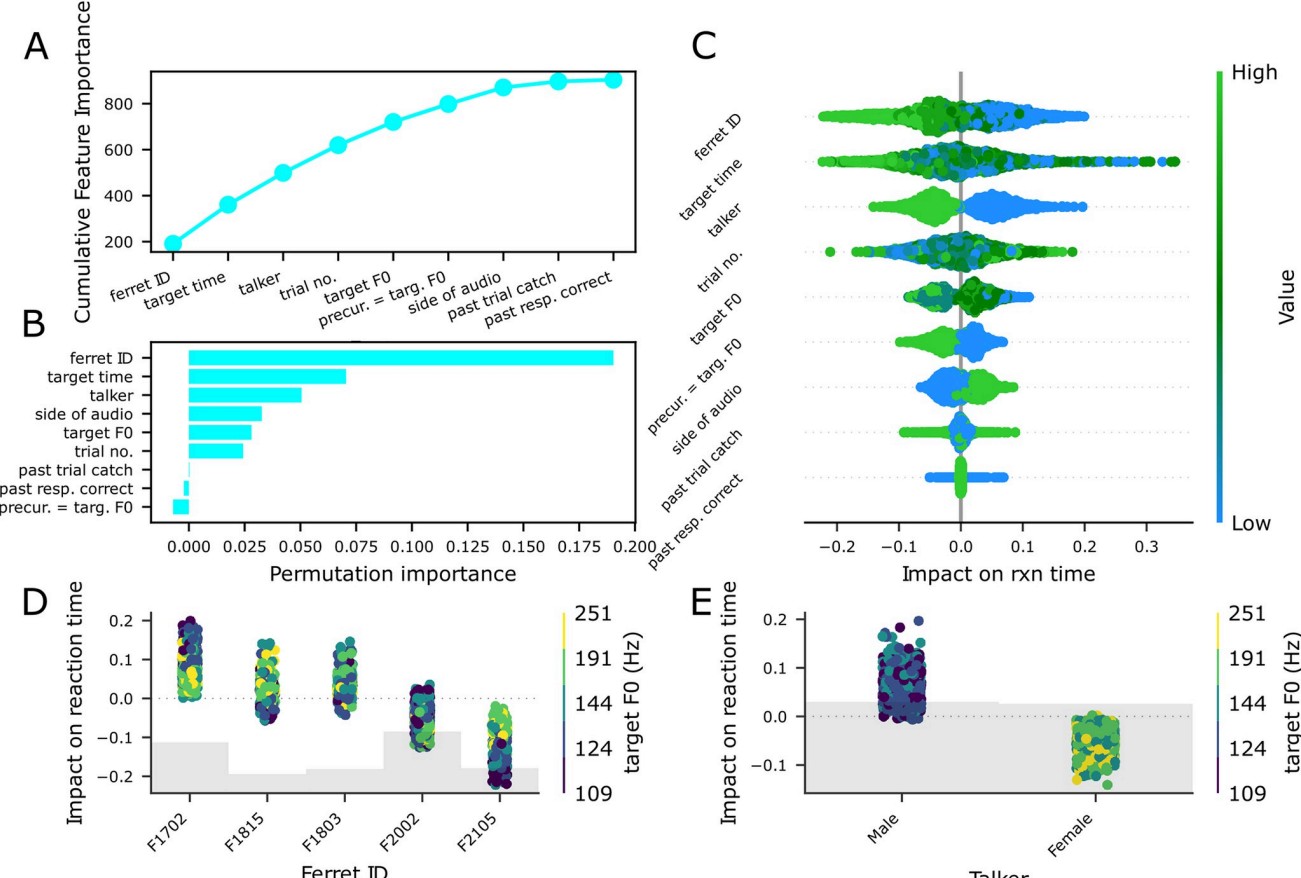

**Fig 4. Reaction time models establish a contribution of F0 to target detection.** A, feature importances of the hit model; B, permutation feature importance of each factor in the model; C, SHAP summary plot of ranked feature SHAP values of each factor in the reaction time model; D, partial dependency plot of SHAP impact versus ferret ID color-coded by target F0; E, partial dependency plot of SHAP impact over talker identity color-coded by the target F0.

responses (Fig 4B), 3/5 ferrets showed this effect, 1/5 had faster reaction times for earlier targets, and 1 showed no difference, S4(A) Fig).

Other factors that significantly predicted reaction times were the side of the audio (left responses were slightly faster than right responses in 2/5 ferrets, right faster than left in 2/5 ferrets, 1/5 did not differ, S4(B) Fig. The model dissociated the effects of talker and F0, with the effect of F0 being somewhat variable across ferrets, with three ferrets showing slower reaction times for the lowest male talker F0, one showing slower reaction times for the pitch-shifted F0 values, and one not showing any F0 effects (Fig 4C). Reaction times were faster when the preceding non-target word had the same F0 as the target in 4/5 animals (S4(C) Fig). Factors that did not influence reaction times—as assessed by the permutation test and feature importance values were the trial number and trial history factors (the previous trial was a catch trial / correct). Therefore, despite equivalent performance in inter and intra-trial roving trials, by applying gradient-boosted regression to the reaction time data, we observe that ferrets' reaction times are faster when pitch provides a consistent streaming cue (Fig 4B and 4E).

## Gradient boosted regression tree models reveal some words elicit more frequent false alarms

Our false alarm model implied that false alarms were potentially lapses in concentration related more to timing than acoustic parameters. However, an alternative possibility is that particular words drive false alarms independently of the characteristics of the talker. To investigate this, we used gradient-boosted regression to ask whether subjects consistently false alarmed to particular non-target words by modeling the animals' response time within a trial based on the word token. We modeled data from the female talker and the male talker separately using only the timing of each word token in a trial, relative to the onset of the trial, to predict the animals' eventual response time (again relative to the onset of the trial rather than the onset of the target word as in the previous reaction time analysis). The prediction accuracy of this model was excellent for both talker types, with a test mse of 0.0193s for the female talker compared to a noise-floor test mse of 1.804s (see Methods) and a train mse of 0.0189s compared to a noise-floor train mse of 1.792s. The test mse for the male talker was 0.0499s compared to a noise-floor test mse of 1.959s, with a train mse of 0.0493s compared to a noise-floor mse of 1.949s (5-fold cross-validation for both train and test metrics).

Reassuringly, in both male and female talker models, the presence and timing of the target word had the strongest predictive power about when animals would release from the center port (Fig 5A–5D). Nonetheless, some words consistently elicited behavioral responses as shown by both feature importance and permutation importance metrics, suggesting that false alarms are not simply temporary lapses in attention but rather that some words are perceived as more similar to the target. Running models on each animal separately (Fig 5E and 5F) confirmed that these were repeatable errors across ferrets and talkers. To better understand the model output, we asked whether any particular acoustic features predicted the errors the animals made.

## Words tokens that elicit false alarms share spectrotemporal similarity with the target

To explore the acoustic features that might underlie the animals' false alarm pattern, we considered three types of measures; first, we used a cochleagram model to estimate the representation of each token at the auditory periphery (Fig 6A–6D), [15], with the caveat that this is a human model, and therefore likely overestimates the frequency resolution available to the

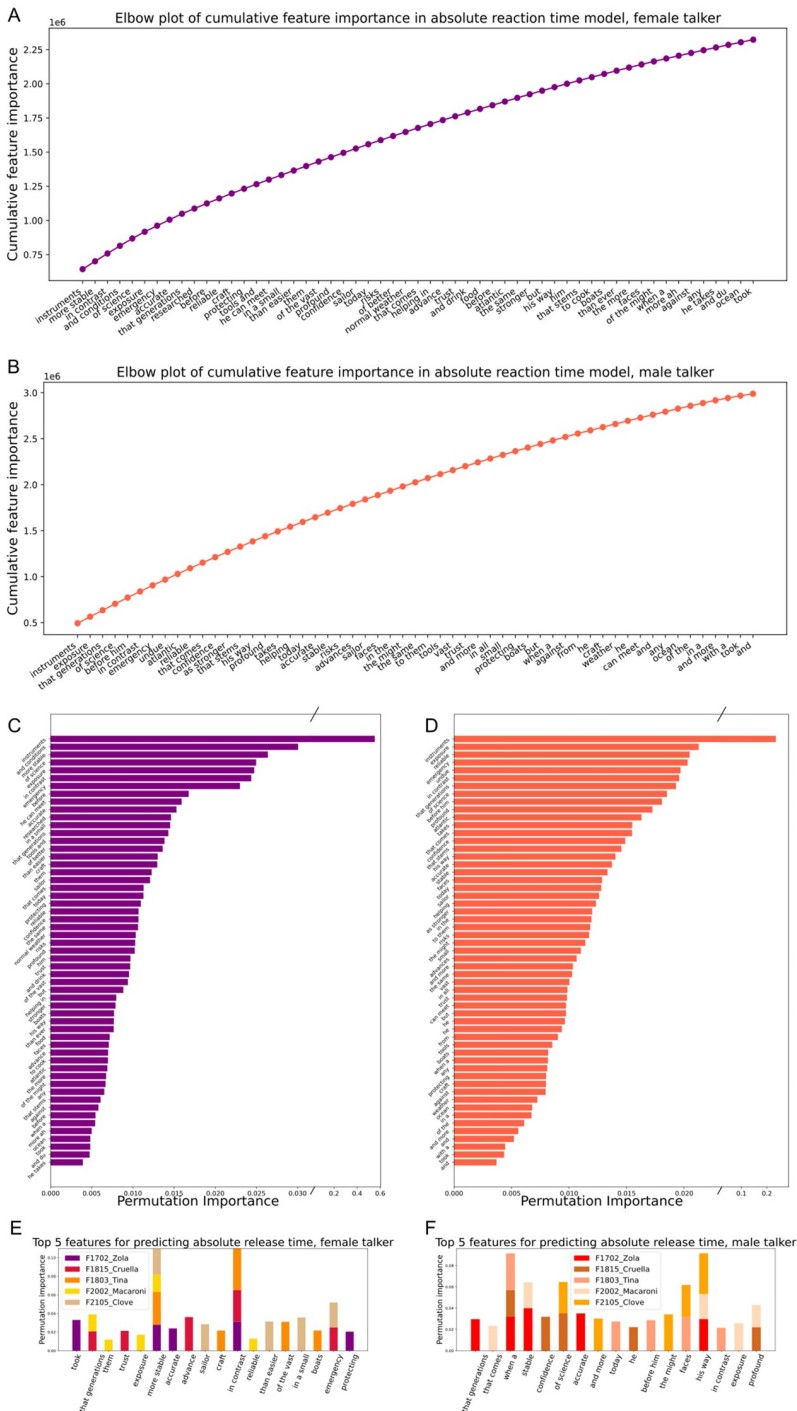

**Fig 5. Gradient boosted models identify words that animals consistently false alarm to.** A, elbow plot of cumulative feature importance in the female talker model; B, same as A but for the male talker; C permutation importance of features included in the female talker model; D, same as C but for the male talker; E, top 5 permutation importances for each individual animal model for the female talker model; F, same as E but for the male talker.

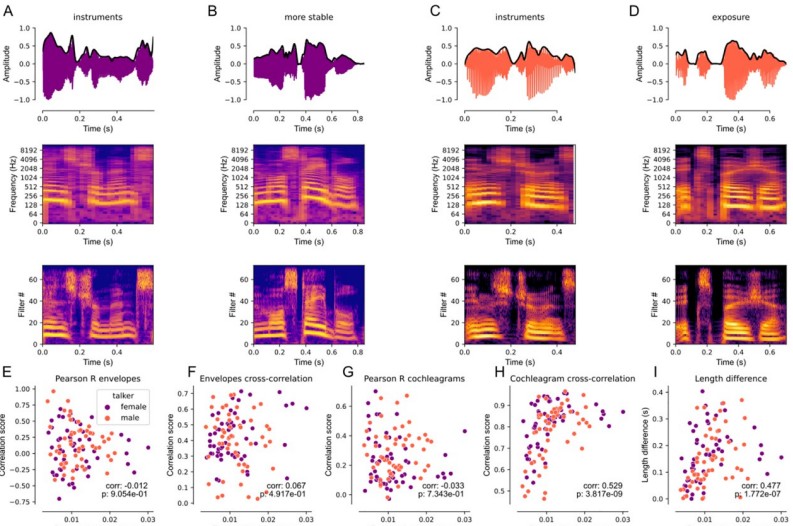

**Fig 6. Spectrotemporal similarity predicts false alarm likelihood.** A, top to bottom: waveform, spectrogram, and cochleagram of instruments for the female talker stimulus. The black line in the waveform plot indicates the extracted envelope. B, top to bottom: waveform, spectrogram, and cochleagram of 'more stable', one of the words associated with a high chance of response in our absolute reaction time model for the female talker. C, same as A but for the male talker stimulus. D, same as B but for the word 'exposure', which was associated with a high rate of response in our male talker absolute reaction time model. E, the Pearson's correlation between the envelopes of the non-target words relative to the target over each non-target word's respective permutation importance. F, the maximum cross-correlation coefficient between each non-target word and the target word over each non-target word's respective permutation importance. G, same as E but using the cochleagram representations of the target and non-target words rather than the envelopes. H, same as F but for the cochleagram of each non-target word relative to the target word rather than the envelope. I, the absolute difference in duration (length) between each non-target and target word over its respective permutation importance.

ferrets). Second, we extracted the envelope of the amplitude waveform in order to explore the role of the temporal envelope. Third, we considered the difference in the duration of each word token and the target word. For the first and second measures, we compared the target and each word token (for all tokens from the same talker) using, firstly, a point-by-point Pearson's correlation, aligning the tokens at their onset. We also calculated the maximum of the cross-correlation to acknowledge that we don't *a priori* know which elements of a given token animals might confuse (e.g. we might imagine the "idence" of "confidence" might be more readily confused with "instr" of "instruments" than "con" might be).

To relate acoustic and behavioral measures, we calculated Spearman's correlation coefficient between the permutation importance derived from the GBMs and each measure of acoustic similarity. The maximum cross-correlation between the cochleagram provided the strongest relationship (Fig 6G spearman's r = 0.529), explaining 28% of the variance in the animals' behavior. Differences in word duration also had a significant relationship with permutation importance (r = 0.424). However, this relationship is in the opposite direction of that that would be predicted if animals were using word duration as a way to identify the target; words with similar durations were associated with a smaller likelihood of a false alarm. Moreover, words with the highest permutation importance can be seen to span a range of duration differences, further confirming the observation that, in all likelihood, similar duration is not a cue that the ferrets are relying upon to solve the task. Neither of the amplitude waveform measures produced statistically significant relationships. From this, we therefore conclude that animals rely most heavily on spectrotemporal features of the word to perform the task.

## Comparison to generalised linear mixed effect models

To compare the average performance of our gradient-boosted tree models with traditional statistical approaches, we used generalised linear mixed effects models (GLMMs) on the variables used in our corresponding gradient-boosted regression tree models with ferret as a random (group) effect. Like our gradient-boosted trees, we implemented five-fold cross-validation for a fair comparison. In most cases, our gradient-boosted decision and regression tree models were comparable to or better than the GLMM approach in terms of their model accuracy. Reassuringly, many of the same statistical main effects were found with both approaches. However, there were some specific instances in which the GBM approach was superior (described below), and the available tools for the visualization of partial dependencies offered the advantage that the non-linear interactions between features could be meaningfully explored and quantified using SHAP values.

A binomial GLMM predicting hit vs. misses had an accuracy of 65.03% for the train split, 64.22% for the test split (this was comparable to the gradient-boosted regression trees, where the model had an average train balanced accuracy of 62.17% and an average test balanced accuracy of 62.09%). Mirroring the GBM, significant coefficients were returned for a talker, audio side, and target pitch (for 191 Hz vs reference of 109 Hz, Fig 7A). Neither the ferret ID, trial number nor target time parameters returned by the GBM were returned as significant by the GLMM (S7 and S8 Tables).

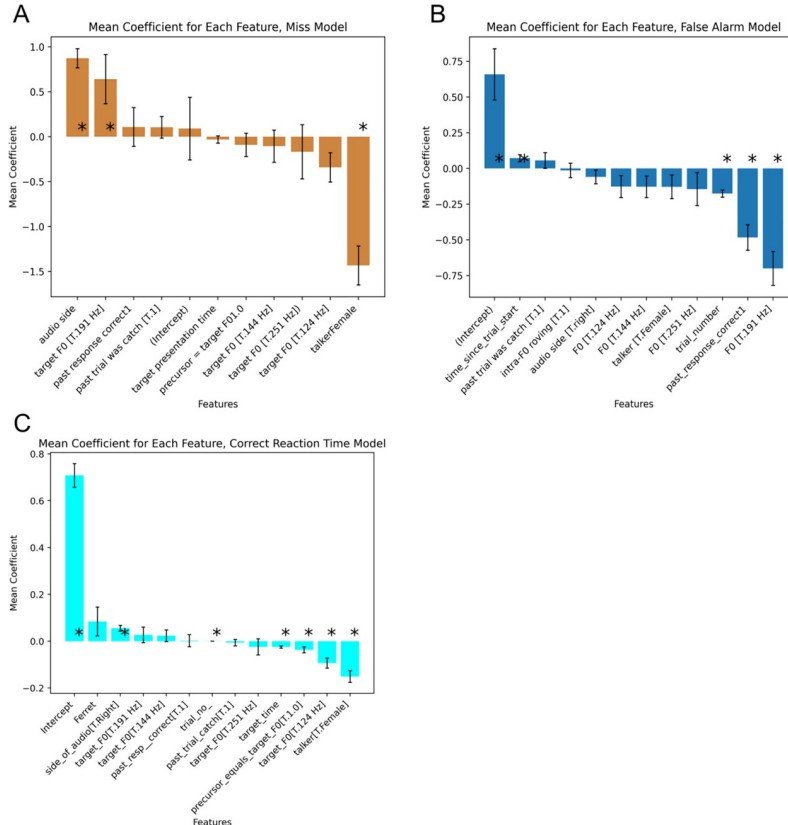

**Fig 7. Mixed effects models show equivalent or worse performance.** Average coefficient values for the mixed effects model predicting A, a miss response for a target trial, B, a false alarm for a catch trial, and C, the reaction time during a correct target trial. Reference talker: male talker, reference F0: 109 Hz, reference side of audio: left side. Asterisks represent mean p-values < 0.05. Error bars represent the mean standard deviation.

A binomial GLMM predicting false alarms during catch trials had lower accuracy than the corresponding GMB model (balanced accuracy was 56.94% vs. 61.5% on the train data set, and 56.50% on the test dataset compared to 61.46%). The mixed effects GLMM returned significant coefficients for the timing variables (time since the trial start, trial number and past response was correct), as well as for the F0 of 191Hz vs. reference 109Hz (Fig 7B, S9 and S10 Tables). The GBM additionally assigned feature importance to ferret ID, and whether the trial was intra-trial roving (Fig 3).

A linear mixed effects model predicting the reaction time for correct hit responses from behavioural variables had a mse of 0.091s for the train dataset and 0.092s for the test dataset, which was comparable to the mse of the gradient-boosted regression tree model (train mse = 0.092s, test mse = 0.102s). Given the restriction that reaction times are between 0–2 s (meaning there are few outliers and a relatively normal distribution), the similarity in performance between the two approaches is perhaps not surprising. The mixed effects model recapitulated the effects of the GBM, returning significant coefficients for talker (faster to female voice), F0 (124 Hz faster than 109 Hz), trials in which the precursor and target had the same F0 were faster than those in which they differed, reaction times were faster for targets later in the trial and for later trials in the session (Fig 7C, S11 and S12 Tables). While the key results were the same across analysis approaches, the ability to visualize SHAP scores for all observations from each animal across multiple variables still provides additional clarity, which could be advantageous when trying to relate brain and behavior. For example, Fig 4D shows how target F0 impacts reaction time for each individual ferret, showing opposite patterns in F1702 and F2105, something that would not be apparent with the mixed effects model coefficients.

Where the GBM excelled was in predicting the absolute release time solely based on which words were in a trial. To match the GBM approach, we used ordinary least squares (OLS) regression, which, like the GBM, did not consider ferret as a factor, and again separated male talker and female talker trials to generate two models. The mse for the OLS model was nearly an order of magnitude larger than the GBM model, 0.15 and 0.19s, respectively for the male and female talker models, compared to errors of 0.0193s and 0.049s for the female and male talker for the GBM (S5(C) Fig, S13 and S14 Tables). Critically, the size of the coefficient for 'instruments' was barely greater than for the first-ranked non-target word in either model. Although there was some similarity in the ranking of non-target words between the linear regression and the GBM, the low overall model accuracy would make it hard to confidently make conclusions about false alarm behavior based on the linear regression alone. This analysis highlights that the GMB model has an advantage when predicting outlier behavior; false alarms to individual non-target word tokens are inherently rare in trained animals, and there is not a fixed response latency (as shown in the reaction time analysis) even if we can assume that animals trigger responses to the onset of word tokens (which the strong relationship between false alarms and cochleagram cross-correlation but not between correlation coefficients suggests is not the case). When performing the response time analysis with the GBM, we subsampled data to ensure that word frequency could not erroneously bias the resulting models; however we repeated the modeling with the original (non-uniform) distribution of word frequencies and the resulting permutation importance scores for non-target words were highly correlated (S5(C) Fig, Spearman's R = 0.72 and 0.87 for female and male talker models respectively) suggesting that this subsampling was unnecessary.

## Discussion

We describe a novel behavioral task in which animals are trained to recognize a target word embedded in a series of non-target words and employed gradient-boosted models to analyze

the subsequent behavior. The results of these models allowed us to understand that, like humans, ferrets are able to form F0-tolerant representations of auditory objects and use F0 to link sounds together into auditory streams. [16, 17]. The ability to identify and discriminate sounds across pitch is likely to be a fundamental property of mammalian audition, as the pitch of a vocal call conveys information about an individual's size, age, and emotional state [18, 19].

We used gradient-boosted models to analyze the rich behavioral dataset we acquired comprising many 1000s of trials from 5 individual animals. We visualized the features that the models used to make predictions using SHAPs feature importance measures and partial dependency plots. This allowed us to understand not only what independent contributions specific variables made to behavior but also how combinations of variables interacted. We compared the output of the GBM models with traditional mixed effects models, which, in most cases, were similar or slightly worse in overall model accuracy and returned very similar main effects. The GBM approach offered two advantages; firstly, the visualization tools are beneficial for understanding how different animals differentially weigh variables when performing the task (which in turn will be helpful for later relating brain and behavior). In a mixed-effects design, this is possible by fitting random slopes in addition to random intercepts. However, understanding and interpreting interaction effects—particularly between multiple categorical variables—quickly becomes intractable. Secondly, for some datasets, where the underlying relationships are inherently non-linear, and the samples are unbalanced, the GBM approach was much more effective, with eventual mean square error substantially lower than corresponding linear regression models. This, in turn, allowed us to relate false alarm behavior to acoustic features, revealing that spectrotemporal similarity was the strongest predictor of an increased likelihood of a false alarm.

The data presented here, in which pitch made only a minor contribution to overall performance, extends previous behavioral work in animals showing that non-human listeners can generalize across variations in F0 for relatively simple sounds. For example, ferrets trained to discriminate artificial vowel sounds with an F0 of 200 Hz maintain their performance at F0s from 150 to 500Hz [5, 20]. Both rats [21] and zebra finches [22] trained to discriminate human speech sounds can generalize across different talkers who naturally vary in their voice pitch, and marmosets can discriminate pitch-shifted vocalizations [23]. However, not all species show pitch constancy; guinea pigs trained to categorize calls (e.g., chut vs. purr) in a Go/No-Go task struggled to perform the task with F0 shifts of +/- half an octave [24]. In our models, F0 had only a very small effect on the ability of animals to correctly identify a target word (Fig 2) or on their likelihood of making a false alarm (Fig 3) and only modest differences in their reaction times (Fig 4). Together, these results suggest that performance is robust across variations in pitch. Our reaction time models suggest that variation in F0 impacts individual animals differently. One benefit of the models developed in this study is that such individual differences can be explored and potentially taken into account when interpreting and analyzing brain signals.

Our analysis of response time data on false alarm trials identified words that animals consistently false alarmed to. Analysis of the underlying acoustic cues highlighted spectrotemporal similarity as the strongest predictor of the likelihood of a word eliciting a false alarm. Previous work in songbirds has found that songbirds do not require spectral cues to distinguish between ascending or descending tones and only need the temporal features of the sound to identify the tones [25]. Other work in mice has shown that mice could discriminate ultrasonic vocalizations but that vocalizations that were similar to one another were correlated with poorer performance, suggesting that mice also use spectrotemporal properties to categorize vocalizations [26]. Recent behavioral work by Osanki and Wang found that marmosets could also categorize intra-species vocalizations through a similar Go/No-Go task paradigm, in which marmosets had to recognize

a target vocalization in the presence of an alternate reference vocalization by licking a metal feeding tube, and could successfully discriminate the same calls when the mean fundamental frequency was shifted upwards from the original F0 [23]; the authors concluded that the marmosets were using multiple acoustic properties to make their categorization choices.

While speech recognition is robust to variation in voice pitch for non-tonal languages, humans use the pitch of complex sounds to separate simultaneous competing sounds and to link sounds together over time to form auditory 'streams.' Auditory streaming has been studied in many species, including frogs [27], starlings [28, 29] and gerbils [30]. Evidence from birds suggests that avians use similar strategies to humans, with differences in intensity and spatial location used to segregate sounds into streams but a greater tolerance to changes in frequency or timing [31]. Ferrets can also detect the presence of 'mistuning' when a single component of a harmonic complex is shifted in frequency, suggesting that, like humans, harmonicity is a strong grouping cue in animals [32]. However, to our knowledge, no one has assessed whether non-human listeners use the pitch of a complex sound in the formation of auditory streams. The impact of pitch roving in increasing the likelihood of false alarms and slowing reaction times is consistent with ferrets using common pitch to link together sounds over time, offering an advantage for subsequent word recognition. Nonetheless, in the absence of a competing stream of information, we cannot be sure that it is streaming per se or simply that greater changes from word token to token make it a more difficult task. One feature of streaming is that it builds up over time [33], and consistent with streaming occurring, the likelihood of missing a target was higher, and reaction times were significantly longer for trials in which the target was early in the stream compared to those in which it was in the middle or late in the stream. We predict that the impact of removing pitch constancy might be more strongly evident in tasks that require separating competing streams.

Here, we demonstrate that gradient-boosted decision trees have high predictive power even when incorporating highly correlated or very sparsely sampled variables and are ideally suited for unpicking multiple contributing factors to behavior. Moreover, this gradient-boosted regression tree method allows us to be agnostic to how factors in our metadata are related to each other and thus presents an excellent way to conduct both hypothesis-driven and exploratory data analysis to uncover otherwise hidden trends in behavioral data and drive analysis. Overall, these findings from these sensitive and powerful models could inform later behavioral and neural data studies by giving us an idea of which behavioral factors impact decision-making in individual animals.

## Supporting information

**S1 Fig. A, bias across trial conditions and talker types; B, reaction times of each animal for correct responses color-coded by F0 of the target word.**
(EPS)

**S2 Fig. Partial dependency plots for the correct hit response/miss response model.** A; SHAP values over the ferret ID color-coded by target presentation time; B, SHAP values over ferret ID color-coded by the side of audio presentation; C, same as B but color-coded by talker type; D, SHAP values over trial number color-coded by whether the trial had the precursor word F0 equal to the target F0.
(EPS)

**S3 Fig. Partial dependency plots for the correct reject/false alarm model.** A, partial dependency plot depicting the mean SHAP impact over the ferret ID color-coded by time within the trial; B, violin plot of the SHAP value over the ferret ID color-coded by the side of audio

presentation; C, violin plot of the SHAP values over the F0 of the trial color-coded by talker type; D; SHAP partial dependency plots of false alarm likelihood by F0, color-coded by ferret ID; E, SHAP values over the F0 of the stream color-coded by trial number; F, same as E but color-coded by time since the start of the trial; Note that while the 191Hz F0 is associated with a higher false alarm rate, this should be interpreted in the context of the much lower FA rate associated with the female talker. G, violin plot of the SHAP value over ferret ID color-coded by talker type; H, SHAP value over ferret ID color-coded by trial number; I, SHAP value over trial duration color-coded by F0.
(EPS)

**S4 Fig. Correct hit response reaction time model partial dependency plots.** A, SHAP values over the ferret ID color-coded by the time to target presentation; B, violin plot of the SHAP value over ferret ID color-coded by the side of audio presentation; C, same as B but color-coded by whether the precursor word's F0 was the same as the target word's F0; D, same as C but color-coded by the talker type for the trial.
(EPS)

**S5 Fig.** A, distribution of the probability of occurrence in the resampled dataset used for the response time model in Fig 5; B, same as A but for the male talker absolute reaction time model; C, scatter plot of the permutation importance of each word with subsampling to equalize the frequency to the distribution plotted in A and B, versus the corresponding permutation importance scores obtained from a model with the uncorrected word distributions.
(EPS)

**S6 Fig. Average coefficients for the mixed effects model predicting the absolute reaction time of A, the female talker and B, the male talker.** Asterisks represent mean p-values < 0.05. Error bars represent standard deviation.
(EPS)

**S7 Fig. Average coefficients for the ordinary-least squares (OLS) model predicting the absolute reaction time of A, the female talker and B, the male talker.** Asterisks represent mean p-values < 0.05. Error bars represent standard deviation.
(EPS)

**S1 Table. Repeated-measures ANOVA for the hit statistic with roving type and talker as factors.**
(PDF)

**S2 Table. Pairwise Tukey HSD posthoc test statistics for the hit statistic comparing the roving types for each talker type.**
(PDF)

**S3 Table. Repeated-measures ANOVA for the false alarm statistic with roving type and talker as factors.**
(PDF)

**S4 Table. Pairwise Tukey HSD posthoc test statistics for the false alarm statistic comparing the roving types for each talker type.**
(PDF)

**S5 Table. Repeated-measures ANOVA for the d' statistic with roving type and talker as factors.**
(PDF)

**S6 Table. Pairwise Tukey HSD posthoc test statistics for the d' statistic comparing the roving type.**
(PDF)

**S7 Table. Average coefficients of the main fixed effects of the miss/correct response model.**
(PDF)

**S8 Table. Average coefficients of the random effects for the miss/correct response model.**
(PDF)

**S9 Table. Average fixed effect coefficients for the false alarm linear mixed effects model.** 1 indicates yes, 0 indicates no.
(PDF)

**S10 Table. Average random effect coefficients for the false alarm linear mixed effects model.**
(PDF)

**S11 Table. Average fixed effect coefficients for the reaction time linear mixed effects model for correct hit responses.** 1 indicates yes, and 0 indicates no.
(PDF)

**S12 Table. Average random effect coefficients mixed effects model predicting reaction time for correct target trial responses.**
(PDF)

**S13 Table. Coefficients for the ordinary least squares (OLS) model predicting absolute reaction time based on word identity in a trial, female talker model.**
(PDF)

**S14 Table. Coefficients for the ordinary least squares (OLS) model predicting absolute reaction time based on word identity in a trial, male talker model.**
(PDF)

**S15 Table. Trial type numbers distributed by ferret ID and talker type (M = male talker, F = female talker).**
(PDF)

**S16 Table. Hyperparameter values for the false alarm categorical model.**
(PDF)

**S17 Table. Hyperparameters for the miss/hit gradient-boosted decision tree model.**
(PDF)

**S18 Table. Hyperparameters for the reaction time gradient-boosted regression tree model predicting the reaction time from the onset of the target word from the subset of correct hit responses.**
(PDF)

**S19 Table. Hyperparameters for the absolute reaction time gradient-boosted regression tree model that predicts the reaction time relative to the female talker type trial start time.**
(PDF)

**S20 Table. Hyperparameters for the absolute reaction time gradient-boosted regression tree model that predicts the reaction time relative to the male talker type trial start time.**
(PDF)

**S21 Table. Hyperparameters for the absolute reaction time models for each ferret ID broken down by Female/Male talker type.**
(PDF)

## Acknowledgments

We thank the hard work and dedication of the Royal Veterinary College technicians and veterinary staff for supporting our scientific experiments and maintaining our animal colony.

## Author Contributions

**Conceptualization:** Carla S. Griffiths, Jennifer K. Bizley.

**Data curation:** Carla S. Griffiths, Jules M. Lebert.

**Formal analysis:** Carla S. Griffiths, Jules M. Lebert.

**Funding acquisition:** Jennifer K. Bizley.

**Investigation:** Carla S. Griffiths, Jules M. Lebert, Joseph Sollini, Jennifer K. Bizley.

**Methodology:** Carla S. Griffiths, Joseph Sollini, Jennifer K. Bizley.

**Project administration:** Jennifer K. Bizley.

**Resources:** Joseph Sollini, Jennifer K. Bizley.

**Software:** Jules M. Lebert, Jennifer K. Bizley.

**Supervision:** Joseph Sollini, Jennifer K. Bizley.

**Visualization:** Carla S. Griffiths, Jules M. Lebert.

**Writing – original draft:** Carla S. Griffiths, Jennifer K. Bizley.

**Writing – review & editing:** Carla S. Griffiths, Jules M. Lebert, Jennifer K. Bizley.

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
