## [Decision Letter · Decision Letter 0]

1 Nov 2023

Dear Dr Bizley,

Thank you very much for submitting your manuscript "Gradient boosted decision trees reveal nuances of auditory discrimination behaviour" for consideration at PLOS Computational Biology.

As with all papers reviewed by the journal, your manuscript was reviewed by members of the editorial board and by several independent reviewers. In light of the reviews (below this email), we would like to invite the resubmission of a significantly-revised version that takes into account the reviewers' comments.

Dear Authors,

As you will see the reviewers found your work interesting but also have raised serious concerns that I would like you to address. Both reviewers 1 and 3, for example, clearly spell out that you have not given GLM a fair chance at competing with your Gradient Boosted Decisions trees. I will look forward to a more detailed comparison.

Best,

Frederic Theunissen

We cannot make any decision about publication until we have seen the revised manuscript and your response to the reviewers' comments. Your revised manuscript is also likely to be sent to reviewers for further evaluation.

Sincerely,

Frédéric E. Theunissen

Academic Editor

PLOS Computational Biology

Daniele Marinazzo

Section Editor

PLOS Computational Biology

Dear Authors,

As you will see the reviewers found your work interesting but also have raised serious concerns that I would like you to address. Both reviewers 1 and 3, for example, clearly spell out that you have not given GLM a fair chance at competing with your Gradient Boosted Decisions trees. I will look forward to a more detailed comparison.

Best,

Frederic Theunissen

Reviewer's Responses to Questions

**Comments to the Authors:**

Reviewer #1: In this study, Griffiths et al. Use a machine learning algorithm, gradient boosted trees, to model behavioral data from ferrets performing a word detection task at a trial-wise level. They both trial outcomes and response times of ferrets’ responses to natural and F0 shifted speech with reasonable success. Importantly, they use ML tools to interpret these models to determine the relative importance of various parameters from behavioral performance. This leads to both confirmatory and novel insights.

Overall, this is a well-executed study that tackles a subject that has received scant attention in the auditory field – detailed modeling of behavioral responses. More generally, these modeling techniques can be used across modalities and species, and should be of wide interest to behavioral neuroscientists. While I overall support the eventual publication of this study, I have a few suggestions for improvement that I hope the authors will find constructive.

Major comments:

1) The authors consider a number of factors affecting performance, but I was surprised to not see temporal factors considered, e.g. the duration of the word, number of syllables, or envelope similarity. These may help explain why the animals are able to maintain performance across varying F0s (if target duration is significantly different from non-target durations) or conversely, why they false-alarm more to specific words (similar durations/envelopes?). Considering the primacy of temporal cues in speech perception, this would be worth modeling!

2) In Fig. 1D, are the labels for inter- and intra-trial F0 changes correct? It appears that inter-trial F0 changes induce both lower p(correct) and higher p(FA) leading to decreased performance (which may not be significant), suggesting that pitch streaming is not providing a behavioral advantage. This seems somewhat contradictory to the claim that there is an advantage to pitch streaming in the abstract and elsewhere. Faster RTs may be the case (Page 6) but this is not necessarily advantageous if performance is decreasing.

3) Writing in abstract and introduction: The title of the paper suggests that the paper is going to be about using ML to model behavioral data. But the beginnings of 1) the abstract emphasizes pitch perception, 2) the summary on hearing loss and brain mechanisms, and 3) the introduction on auditory cortex and inactivation studies. Overall, this results in a confusing entry into the story. I would suggest that the authors reframe these sections by focusing on the need to model behavioral data, other models that are out there (e.g., drift diffusion models or LATER models), what ML approaches can offer etc. The details can of course be there, but not having them in stress positions would help alleviate this confusion.

4) The paper is highly technical and not easily accessible to biologists. Some explanation is provided in the methods, but I would have liked a more didactic paragraph in the introduction on gradient boosted trees and SHAP plots, and some help interpreting these plots in the Results section.

Minor:

Fig.1: Could you use different colours to signify the trial types in Figs. 1D-G and F0 value in Fig. 1H?

Figs 2 -5: Could you please standardze the font size used for labelling axes? Some are too small to be readable.

Page 10 Line 318 Please provide a reference or description of the STRAIGHT pitch shifter

Additional citations featuring behavioral responses in animals to F0 shifts: Osmanski and Wang, PNAS 2023, Kar et al., eLife, 2022 (also includes a model), Bregman et al., PNAS 2016, Neilans et al. Plos ONE 2014.

Reviewer #2: This manuscript develops a novel application of a machine learning tool to the analysis of complex auditory behavior. The authors use gradient boosted decision trees and -regression to analyze data from ferrets during a word detection task, in which the timing, location, speaker and pitch of target and distractor words varies across trials and sessions. The analyses provide a quantitative measure of the relative importance of different variables to hit/miss, false alarm, and reaction time outcomes, confirming the impact of some variables identified using more traditional analysis. The analysis supports the authors’ hypothesis that ferrets are mostly able to generalize categorical representation of words across different voice pitch. There are some other interesting observations, e.g., speaker identity influences hit rate but not false alarm rate.

This work takes a novel approach to analyzing an interesting new behavior. The authors make a reasonable case that gradient boosted tree methods address limitations of more traditional methods for analyzing behavior, and they demonstrate that these methods can be applied to their behavioral data. However, the novelty of the method and the specific questions about behavior seem muddled, raising questions about the value of this fairly complex analysis. These are probably addressable--largely by addressing question around interpretation of effect size in the new models’ results and a meaningful comparison with simpler, traditional methods.

MAJOR CONCERNS

1. What is this manuscript about? Reading the title, one might imagine that it is intended to validate a new method for analyzing behavior. But reading the introduction and discussion, it sounds like gradient boosted decision trees are established, and the questions are focused on specific variables related to a specific aspect of auditory processing. The results are somewhere in the middle, emphasizing the novelty of gradient boosted trees, but not really demonstrating when they are able to show things that more traditional methods of behavioral analysis don’t show. In the Methods, the authors cite “Linear mixed effect and generalized linear models” (L. 353) as established approaches. The baseline analysis in Fig. 1 seems much simpler than these norms, only considering one or two variables at a time and not really pushing the traditional methods. A simple linear classifier could be trained to predict outcomes like hit vs. miss, and it seems like the weight of each variable in the classifier could be compared directly to the decision tree statistics. Without this comparison, a reader finds themselves wondering why they use the more complex and opaque method proposed here. It would help if either the authors could provide a quantitative contrast with a traditional method OR rewrite some key parts of the manuscript to make it clear that the approach does not need validation.

2. “feature importance, … SHAP feature values” (L 115.). “SHAP feature importances” (Fig. 2B) “SHAP partial dependency” (Fig. 2C). It’s clear that the relative size of these statistics provides a measure of how much the corresponding variables are able to explain behavioral outcomes. However, the units and magnitude of the various effects are not obvious. Digging into the methods, the reader learns that these are “Shapley Additive Importance features” (L. 408) but not much else. To be more specific: What does a step from 4500 to 5000 in Fig. 2A mean? Why is the scale to 55000 in Fig. 3A? Can an x value in Fig. 2B be interpreted as a probability? What are the shaded boxes in Fig. 2C? Or the meaning of the individual dots? (single trials?) The words in the text provide a high-level interpretation, but a better explication of the numbers is needed. A place to start would be to define statistics in the Results and provide meaningful units for the plots. Ideally, these would translate to human-interpretable units that give a sense of the magnitude of impact on behavior outcomes. In more traditional linear analysis, a useful concept, for example, is variance explained. That may or may not be possible with the method used here, but some explication seems necessary.

2a. Currently each analysis gives a list of results, but it is hard to get a sense of the “nuances” revealed by the three different models. Something that might be helpful would be a comparison of relative effect size/importance for each variable between the models. Eg, could a scatter plot contrast the effect of all the model variables (talker identity, animal, side, pitch shift type, etc.) on hit/miss vs. their effect on FA rate? If not a new figure, some of side-by-side discussion would be helpful. The authors do point out that the same variables cannot be included in each model, so it makes sense that a complete direct comparison may not be possible.

LESSER CONCERNS

(L. 46) “20,487 trials” It would be helpful to indicate the number of animals here too.

(L. 56) What was the range of trial lengths, i.e., distribution of time from trial start to target onset?

(L. 60) “Single male and single female” Does “single” mean one value of F0 per voice?

(L. 61) “trials were introduced in which the F0…” Can the authors provide a sense of how large a portion of trials were used to probe pitch generalization? It’s not clear if these trials were probes in the traditional sense or if the animals could learn that multiple different pitches could be target sounds. Did performance on the variable pitch trials change over time?

(L. 79) “trained F0 values” What are the trained F0 values? Not clear in the figure (1H?)

(L. 108, L. 136) The choice of what independent variables to include in which analysis is not always clearly motivated. Is “precursor=target F0” (Fig 2) different from “intra-trial F0 roving” (Fig. 3)? Or are they the same thing? It would help to identify what variables are shared across models and what variables are (out of necessity) different.

(L. 115) “SHAP” please define here.

(L. 165) “mean squared error ... was 0.0947” What is the baseline error, e.g., for a completely random model? Can this be translated into percent correct, as for the hit/miss model?

(L. 168) “whether the F0 of the previous word equaled the target word” Is this the same as inter- vs. intra-trial roving?

(L. 196). “test mse…” Again (as L. 165), can a noise floor be provided? Understandably, percent correct doesn’t make sense here.

(L. 204) “The commonly false alarmed words…” A cursory inspection of spectrograms of the high-FA words suggests that they do share structure with the target word. Can the authors perform a comparison between spectrograms? For example, one could compute the correlation, point-by-point between the target and confused word spectrogram for the same voice.

(L. 269) What was the overall sound level of the stimuli?

(L. 306) “3/5 animals…” did training history predict animal-specific effects in the model analysis?

(L. 383) The authors appear to have carefully controlled sampling to avoid potential bias. They then report that results were the same as if they didn’t subsample. Given the non-parametric nature and flexibility of the gradient boosted tree method, is balancing independent variables necessary? One might think that the method is intrinsically able to account for these factors. A reference indicating why or why not these preprocessing steps were necessary would be helpful.

MINOR/TYPO

(L. 28-29) “… frequently … often …” seems redundant.

(L. 97) “they do” should be “it does”?

(L. 99) “parameters of was the…” Confusing grammar

(L. 403) “Figure 5S” should be “Figure S5?

Reviewer #3: This is an interesting MS that offers several contributions, From a behavioral point of view, it demonstrates that ferrets can be trained to categorize and recognize a target word in a stream of other words regardless of its pitch and speaker’s voice. This is always a valuable contribution since such tasks can be used in a variety of studies. The other contribution is the utilization of the gradient boosted decision trees to do the analysis, and to demonstrate its power and utility for the analysis of a truly large set of behavioral data.

The paper is very well written and is quite clear and succinct. I do have several questions, suggestions, and perhaps some qualms regarding the interpretations of the results.

1. The ferrets of course were trained to generalize across pitch and voice in all the long initial phases of the training. The animals were trained extensively o

---

## [Decision Letter · Decision Letter 1]

9 Mar 2024

Dear Dr Bizley,

We are pleased to inform you that your manuscript 'Gradient boosted decision trees reveal nuances of auditory discrimination behavior' has been provisionally accepted for publication in PLOS Computational Biology.

Best regards,

Frédéric E. Theunissen

Academic Editor

PLOS Computational Biology

Daniele Marinazzo

Section Editor

PLOS Computational Biology

Reviewer's Responses to Questions

**Comments to the Authors:**

Reviewer #1: The authors have addressed all concerns arising from the previous round of reviews. I thank them for putting in the effort to educate readers on these powerful behavioral analysis tools. Congratulations on a nice study!

Reviewer #2: The revised manuscript addresses the previous concerns throughly, and it now provides a clear illustration of how the decision tree models complement and expand from previously established approaches to analyzing behavioral data.

Just a few requests/comments, up to the authors/editor, but which might improve accessibility of the findings:

Figure 2 illustrates several related metrics for assessing the importance of different input variables (cumulative feature importance, permutation importance, SHAP feature importance) but the explanation of what distinct insights each of these provide requires a close read of the manuscript. It would help if a bit more context was provided for the information provided by these factors when they are first illustrated.

(L. 183) Criteria for establishing significance of specific factors would also be useful. Why are past resp. correct and past trial catch excluded while target time is not? All the bars look small in Fig 2B.

Grammar/typo? Something is missing (L 190): “The trial number (with trial earlier in the session reducing the likelihood of a miss, and later trials being associated with higher miss rates).”

Reviewer #3: I am impressed very much by the thorough manner in which the authors replied to the reviews, the extensive additional analysis they conducted, and their patience with the myriad requests of the reviewers. Well Done and I recommend publication as is

**Have the authors made all data and (if applicable) computational code underlying the findings in their manuscript fully available?**

Reviewer #1: **No: **Placeholder text is in place to provide link to data upon publication.

Reviewer #2: Yes

Reviewer #3: Yes

PLOS authors have the option to publish the peer review history of their article (what does this mean?). If published, this will include your full peer review and any attached files.

Reviewer #1: No

Reviewer #2: No

Reviewer #3: No

---

## [Editor Report · Acceptance letter]

8 Apr 2024

PCOMPBIOL-D-23-01539R1 

Gradient boosted decision trees reveal nuances of auditory discrimination behavior

Dear Dr Bizley,

I am pleased to inform you that your manuscript has been formally accepted for publication in PLOS Computational Biology. Your manuscript is now with our production department and you will be notified of the publication date in due course.

With kind regards,

Anita Estes
